# Inhibitors of Rho/MRTF/SRF Transcription Pathway Regulate Mitochondrial Function

**DOI:** 10.3390/cells13050392

**Published:** 2024-02-24

**Authors:** Pankaj Patyal, Xiaomin Zhang, Ambika Verma, Gohar Azhar, Jeanne Y. Wei

**Affiliations:** Donald W. Reynolds Department of Geriatrics and Institute on Aging, University of Arkansas for Medical Sciences, Little Rock, AR 72205, USA; ppatyal@uams.edu (P.P.); zhangxiaomin@uams.edu (X.Z.); averma@uams.edu (A.V.); azhargohar@uams.edu (G.A.)

**Keywords:** Rho/MRTF/SRF, mitochondria, bioenergetics, histone-4, acetylation, oxidative stress

## Abstract

RhoA-regulated gene transcription by serum response factor (SRF) and its transcriptional cofactor myocardin-related transcription factors (MRTFs) signaling pathway has emerged as a promising therapeutic target for pharmacological intervention in multiple diseases. Altered mitochondrial metabolism is one of the major hallmarks of cancer, therefore, this upregulation is a vulnerability that can be targeted with Rho/MRTF/SRF inhibitors. Recent advances identified a novel series of oxadiazole-thioether compounds that disrupt the SRF transcription, however, the direct molecular target of these compounds is unclear. Herein, we demonstrate the Rho/MRTF/SRF inhibition mechanism of CCG-203971 and CCG-232601 in normal cell lines of human lung fibroblasts and mouse myoblasts. Further studies investigated the role of these molecules in targeting mitochondrial function. We have shown that these molecules hyperacetylate histone H4K12 and H4K16 and regulate the genes involved in mitochondrial function and dynamics. These small molecule inhibitors regulate mitochondrial function as a compensatory mechanism by repressing oxidative phosphorylation and increasing glycolysis. Our data suggest that these CCG molecules are effective in inhibiting all the complexes of mitochondrial electron transport chains and further inducing oxidative stress. Therefore, our present findings highlight the therapeutic potential of CCG-203971 and CCG-232601, which may prove to be a promising approach to target aberrant bioenergetics.

## 1. Introduction

Recent advances in our understanding of the essential role of mitochondrial metabolism in disease have rapidly expanded and provided new insights for diagnosis and treatment. Most importantly, the key role of mitochondria in cancer development has been extensively explored [1]. The emerging studies suggest that mitochondria are not only important in cancer cell viability, but may also play an essential role in metastasis and tumorigenesis [2,3]. Therefore, mitochondria have become an important organelle and a potential pharmacological target for cancer therapy. 

Mitochondria are the main source of adenosine triphosphate (ATP) through the processes of oxidative phosphorylation (OXPHOS), β-oxidation, and the tricarboxylic acid cycle. They also regulate calcium homeostasis [4,5]. Moreover, mitochondria are the major source of reactive oxygen species (ROS) and control apoptotic cell death. The complexes I and III of the electron transport chain (ETC) within mitochondria can generate ROS that can oxidize proteins, lipids, and nucleic acids [6,7,8]. Mitochondria are highly organized and dynamic organelles that are in continuous states of fission and fusion. They consist of GTPases that divide and fuse the mitochondrial membranes. Mitofusin 1 (Mfn1) and mitofusin 2 (Mfn2) are large GTPases that function in outer membrane fusion while optic atrophy type 1 (Opa1) is required for inner membrane fusion [9,10]. Mitochondrial fission protein 1 (Fis1) and dynamin related protein 1 (Drp1) assemble on the mitochondrial surface to mediate mitochondrial fission [11]. Peroxisome proliferator-activated receptor-γ coactivator (PGC-1α) and its isoform PGC-1β regulate mitochondrial genes involved in biogenesis, including components of the electron transport system [12,13].

The upstream regulation of myocardin-related transcription factors (MRTFs)—serum response factor (SRF) by RhoA—modulates actin dynamics to induce the nuclear accumulation of the MRTF-A and MRTF-B [14]. Various extracellular stimuli can activate Rho GTPases, particularly RhoA, that stimulates actin polymerization. This event of polymerization sequesters MRTFs in the cytoplasm, allowing MRTFs to be released from G-actin and become available for nuclear translocation. These MRTFs interact with SRF, and the MRTF-SRF complex binds to serum response elements in the promoter regions of target genes, thereby activating the transcription of genes [15]. This signaling cascade has been implicated in cancer cell migration, metastasis, and pathological fibrosis [16]. 

RhoA signaling can affect mitochondrial dynamic, function, and distribution within the cell. RhoA activation regulates Drp1-mediated mitochondrial fission and increases mitophagy [17,18]. MRTF-SRF signaling promotes the transcription of genes involved in actin polymerization, and proper actin dynamics are essential for mitochondrial movement and distribution within the cell [19]. SRF is involved in the transcription of genes regulating sarcomere assembly and mitochondrial metabolism [20]. Many ternary complex factors and cofactors have been shown to interact with SRF and induce its transcriptional activation. One of the cofactors reported by our group was p49/SRF-dependent transcription regulation-associated protein (p49/STRAP). It has been shown that a subunit of complex I interacts with p49/STRAP and regulates mitochondrial function [21,22]. Moreover, MRTFs recruit chromatin-remodeling enzymes such as p300 (a histone acetyltransferase) to enhance SRF-mediated target gene expression [23]. Studies have also shown that SRF-regulated chromosomal templates by RhoA induces histone H4 hyperacetylation, and histone H4 regulates the mitochondrial DNA transcription [24,25]. The discovery of a novel series of oxadiazole-thioether CCG-molecules, inhibitors of the Rho/MRTF/SRF transcription pathway, has implicated their role in preventing cancer cell migration in vitro and improving fibrosis in vivo [26,27]. However, the molecular mechanism of their action remains elusive, and whether these small molecule inhibitors regulate mitochondrial function is not clear.

In this study, we investigated the basic mechanism of Rho/MRTF/SRF inhibition signaling using CCG-203971 and CCG-232601 in normal cell lines of human lung fibroblasts and mouse myoblasts. We explored the epigenetic modification of histone H4 acetylation and demonstrated the impact of CCG-203971 and CCG-232601 on mitochondrial dynamics, biogenesis, and overall function. We show these second-generation CCG drugs alter the actin cytoskeleton. Furthermore, we found these small molecule inhibitors induce oxidative stress and reduce the total ATP production. These observations suggest that CCG-203971 and CCG-232601 can be used in preclinical evaluations as novel therapeutic strategies targeting mitochondria.

## 2. Materials and Methods

### 2.1. Small Molecule Inhibitors

CCG-203971 (cat. no. 15075) and CCG-232601 (cat. no. 24382) molecules were purchased from Cayman Chemical, Ann Arbor, Michigan, USA. They were dissolved in dimethylsufoxide (DMSO) and stored at −20 °C.

### 2.2. Cells and Culture Conditions

We purchased the WI-38 human lung fibroblast cell line (American Type Culture Collection [ATCC], Manassas, VA, USA, cat. no. CCL-75) and mouse skeletal muscle C2C12 cell line (ATCC, Manassas, VA, USA, cat. no. 1772). WI-38 cells were cultured in Eagle’s minimum essential medium (ATCC, Manassas, VA, USA, cat. no. 30-2003), while C2C12 cells were cultured in Dulbecco’s Modified Eagle Medium (DMEM) (ThermoFisher Scientific, Waltham, MA, USA, cat. no. 11965092). Both media were supplemented with 10% (*v*/*v*) heat-inactivated fetal bovine serum (ThermoFisher Scientific, Waltham, MA, USA, cat. no. A5256701) and 1% penicillin/streptomycin (ThermoFisher Scientific, Waltham, MA, USA, cat. no. 15140122). All cell lines were grown in Falcon 75 cm^2^ tissue culture flasks (ThermoFisher Scientific, Waltham, MA, USA, cat. no. 13-680-59) and maintained at 37 °C with 5% CO_2_.

### 2.3. Cell Viability Assay

We used the MTS Assay Kit (Abcam, Waltham, MA, USA, cat. no. ab197010) to determine cell viability. Briefly, WI-38 and C2C12 cells were seeded at 2 × 10^3^/well in a 96-well microtiter plate. DMEM treated with different concentrations of CCG-203971 and CCG-232601 was added to a final volume of 200 μL/well and incubated for 24 h; 0.5% DMSO was used as control. After treatment, 20 μL of the MTS reagent (5 mg/mL) was added per well, and the plates were incubated at 37 °C for 3 h in a humidified 5% CO_2_ incubator according to the manufacturer instructions. Absorbance was read at 490 nm with a BioTek Synergy H1 microplate reader (Agilent, Santa Clara, CA, USA). The lack of (or reduced) colored MTS tetrazolium/formazan indicated cytotoxicity. IC_50_ values were determined by non-linear regression analysis using GraphPad Prism 8.4.3. 

### 2.4. Immunofluorescence Staining

We seeded WI-38 and C2C12 cells at 1.0 × 10^5^ cells per well in a 35 mm MatTek microscopy glass dish (ThermoFisher Scientific, Waltham, MA, USA, cat. no. NC9574048) and treated with 20 μM of CCG-203971 and 20 μM of CCG-232601 for 24 h; 0.5% DMSO treatment was used as control. Briefly, cells were fixed with 3.7% formaldehyde (ThermoFisher Scientific, Waltham, MA, USA, cat. no. SF100-4) for 10 min and permeabilized with 0.2% Triton X-100 for 5 min. Cells were washed with phosphate buffered saline (PBS) and blocked with 3% bovine serum albumin for 1 h at room temperature and then incubated with rhodamine-phalloidin (1:500) (ThermoFisher Scientific, Waltham, MA, USA, cat. no. R37110) for 20 min. Cells were stained with MitoTracker Red CMXRos (1:1000) (Invitrogen, Waltham, MA, USA, cat. no. M7512) for 30 min. DAPI (1:1000) (ThermoFisher Scientific, Waltham, MA, USA, cat. no. D1306) was used to label the nuclei. Fluorescent images were captured with a Zeiss LSM 880 Confocal Microscope with ZEN 3.2 (blue edition) software (Carl Zeiss Microscopy, White Plains, NY, USA). The average relative fluorescence intensity was measured with the ImageJ (NIH). A one-way ANOVA test was performed to compare MitoTracker labeling in control cells versus treated cells.

### 2.5. Quantitative Real-Time Polymerase Chain Reaction

RNA was isolated with RNeasy Mini Kit (Qiagen, Germantown, MD, USA, cat. no. 74104) and TRIzol reagent (Invitrogen, Waltham, MA, USA, cat. no. 15596026). Isolated RNA was quantified with a spectrophotometer ND-1000 (NanoDrop, Wilmington, Delaware, USA). Reverse transcription to cDNA was performed with the iScript cDNA Synthesis Kit (BioRad, Hercules, CA, USA, cat. no. 1708890) per manufacturer instructions. For qRT-PCR analysis, 1 μg cDNA per reaction was amplified with PowerTrack SYBR Green Master Mix for qPCR (ThermoFisher Scientific, Waltham, MA, USA, cat. no. A46012). Changes in expression were analyzed with the 2^−ΔΔCT^ method and reference to the internal control gene, 5s. The relative expression was calculated by normalizing the change in expression to control for each gene. Primer sequences that were used are described previously [28].

### 2.6. Western Blot Analysis and Immunoprecipitation

We performed Western blot analysis as described previously [29]. After being treated with 20 μM of CCG-203971 and 20 μM of CCG-232601 for 24 h, cells were washed with ice cold PBS and lysed with RIPA lysis buffer (Santa Cruz, Dallas, Texas, USA, cat. no. sc-24948A), followed by clarification of lysates by centrifugation at 12,000 rpm for 20 min at 4 °C. Lysate protein concentrations were determined with the Pierce BCA Protein Assay Kit (ThermoFisher Scientific, Waltham, MA, USA, cat. no. 23227). Total protein (50 µg) was loaded on sodium dodecyl sulfate–polyacrylamide gel electrophoresis (SDS-PAGE) gels and were electroblotted onto nitrocellulose membrane. For immunoprecipitation (IP), dynabeads were conjugated with histone H4 antibody (1 µg/sample) for 1 h followed by washing of beads 3 times with RIPA lysis buffer. Lysates protein (500 µg/sample) from treated or control cells were added into histone H4 conjugated beads and incubated using a shaker at 4 °C for 1 h. Beads were then washed 3 times with same lysis buffer and resuspended in 50 μL of 4× sample buffer, followed by boiling for 10 min and separation by SDS-PAGE (15%). IP blots were developed against anti-acetyl-H4K5, anti-acetyl-H4K8, anti-acetyl-H4K12, and anti-acetyl-H4K16. For loading control H4, 20 µg protein was denatured and loaded on SDS-PAGE from total lysate. Primary antibodies included p49/STRAP (1:1000) antibody [30], SRF (1:1000, cat. no. SC-335), PGC-1α (1:1000, cat. no. sc-518025), and GAPDH (1:1000, cat. no. sc-365062) purchased from Santa Cruz. RhoA (1:1000, cat. no. 2117), MRTF-A (1:1000, cat. no. 97109), MRTF-B (1:1000, cat. no. 14613), histone H4 (1:1000, cat. no. L64C1), anti-acetyl-H4, Lys16 (1:1000, cat. no. E2B8W), anti-acetyl-H4, Lys12 (1:1000, cat. no. D2W6O), anti-acetyl-H4, Lys5 (1:1000, cat. no. D12B3), anti-acetyl-H4, Lys8 (1:1000, cat. no. 2594) purchased from cell signaling (Danvers, MA, USA) and OXPHOS in 1:5000 (ThermoFisher Scientific, Waltham, MA, USA, cat. no. 45-8099). The secondary antibodies included anti-mouse HRP in 1:5000 (Invitrogen, Carlsbad CA, USA, cat. no. 62-6520), anti-goat HRP in 1:5000 (Santa Cruz, Dallas, Texas, USA, cat no. sc-20200), and anti-rabbit AP in 1:5000 (Bio-Rad, Hercules, CA, USA, cat. no. 64251130). Immunoreactive bands were visualized with ECL and iBright CL1500 (Invitrogen, Waltham, MA, USA). Densitometric analysis was conducted with ImageJ software (Version 1.54g, National Institutes of Health). 

### 2.7. Extracellular Flux Assays 

To measure oxidative consumption rate (OCR), extracellular acidification rate (ECAR), and ATP rate, the Seahorse XFe96 Analyzer (Agilent, Santa Clara, CA, USA) was used. Following the manufacturer protocol, equal number of cells (12,000 cells) were seeded per well in a XF Cell Culture Microplate (Agilent, Santa Clara, CA, USA) with complete media at 37 °C, 5% CO_2_, and 100% humidity. After 24 h of treatment with different concentrations of CCG molecules, the cells were washed with Seahorse XF DMEM media (pH 7.4) and left in Seahorse XF DMEM media for 1 h at 37 °C in a non-CO_2_ incubator. The real-time cell metabolic function was measured with the Seahorse XF Cell Mito Stress Test Kit (Agilent, Santa Clara, CA, USA, cat. no. 103015-100) and the Seahorse XF Glycolytic Rate Assay Kit (Agilent, Santa Clara, CA, USA, cat. no. 03344-100) and the Seahorse XF Real-Time ATP Rate Assay Kit (Agilent, Santa Clara, CA, USA, cat. no. 03592-100). Measurements from OCR, ECAR, and ATP assay experiments were normalized to equal number of cells in all variables. After the experiment, seahorse plate was used to perform cell count using trypan blue exclusion method and data is normalized accordingly. 

### 2.8. High Resolution Respiratory Analysis

The Oxygraph-O2k high-resolution respirometer (Oroboros Instruments, Innsbruck, Austria) was used to analyze the mitochondrial function at complex levels. Briefly, 2 × 10^6^ cells were permeabilized with digitonin (8 μM/million cells) in MiRO5 buffer (Oroboros Instruments, Innsbruck, Austria) for 20 min at 4 °C. The detailed method of substrate-inhibitor-titration protocol was described previously [31]. Data were exported and analyzed with DatLab 6.2 software (Oroboros Instruments, Innsbruck, Austria), and cellular respiration of each mitochondrial complex was expressed as oxygen flux (pmol/s*million cells). 

### 2.9. Flow Cytometry

The mitochondrial membrane potential was measured with flow cytometry by staining with JC−1 fluorescent dye (5 µg/mL) (ThermoFisher Scientific, Waltham, MA, USA, cat. no. T3168). Cells were seeded at a density of 1 × 10^6^ cells/well and after 24 h of treatment with 20 µM of CCG molecules, samples were harvested and washed with cold PBS. Pellets were resuspended and samples were incubated for 10 min at room temperature with JC−1 dye. The cells were centrifuged and washed gently 3 times in warm Hank’s balanced salt solution (HBSS) buffer, followed immediately with flow cytometry analysis. Mitochondrial ROS production was measured similarly with MitoSOX Red (ThermoFisher Scientific, Waltham, MA, USA, cat. no. M36008). Samples were stained for 20 min in 37 °C non-CO_2_ incubator. After washing with HBSS, cells were sorted with LSRII Flow Cytometer (BD Biosciences, San Jose, CA, USA) and further analyzed by FlowJo_v10.8.1 software (BD Biosciences, San Jose, CA, USA). 

### 2.10. Statistical Analysis

Data were presented as mean ± SD in which n represents the number of independent replicates. Statistical analyses were conducted with Prism 8.4.3 (GraphPad). Differences between multiple comparisons were analyzed with one-way analyses of variance (ANOVA) followed by Tukey’s test. The experimental groups were considered significantly different at * *p* < 0.05, ** *p* < 0.01, *** *p* < 0.001, **** *p* < 0.0001.

## 3. Results

### 3.1. CCG-203971 and CCG-232601 Exhibit Dose Dependent Cytotoxicity against WI-38 and C2C12 Cell Lines 

To assess the cytotoxic activity of CCG molecules, we performed MTS Assay on WI-38 human fibroblast and C2C12 mouse myoblast cell lines. The data is shown as a percentage of cell viability with respect to the control cells (0.5% DMSO vehicle treated). WI-38 cells (Figure 1) and C2C12 cells (Appendix A) were exposed to 0, 10, 100, and 500 nM, and 10, 20, 50, and 100 µM of both CCG-203971 and CCG-232601 for 24 h. At low doses of CCG molecules, there was no effect on the cellular viability of cells in either cell line. However, at a high dose of 50 µM, cell viability decreased to 70%, and at 100 µM, cell viability decreased to 50% in WI-38 cells. With a dose of 50 µM in C2C12 cells, cell viability decreased to 80%, and at 100 µM, it decreased to 60%. This indicates that WI-38 cells are more sensitive to the CCG molecules when compared to C2C12 cells. Furthermore, CCG-203971 and CCG-232601 both exhibit similar effect on cell viability. The data, shown as IC_50_ ± SD, are presented in Figure 1 and in Appendix A. CCG-203971 and CCG-232601 are equipotent towards both WI-38 (IC_50_ = 12.0 ± 3.99 µM and 14.2 ± 2.57 µM, respectively) and C2C12 cells (IC_50_ = 10.9 ± 3.52 µM and 12.9 ± 2.84 µM, respectively).

### 3.2. CCG-203971 and CCG-232601 Specifically Inhibit Rho/MRTF/SRF Signaling Pathway 

The regulatory mechanism of SRF-mediated gene transcription includes MRTF-A and MRTF-B, which are both regulated by Rho [32]. Despite the fact that CCG molecules are pleiotropic, the mechanism of inhibition of RhoA transcriptional signaling is not clear. Therefore, we first sought to explore their mechanism of inhibition of Rho/MRTF/SRF signaling. WI-38 fibroblasts and C2C12 myoblasts were treated with 10 µM and assessed at 24 h via Western blot analysis, but no change was observed in different protein expression levels. Furthermore, a 20 µM dosage treatment was assessed at 24 h. Figure 2 shows that 20 µM CCG molecule treatment resulted in downregulation of RhoA, MRTF-A, MRTF-B, p49/STRAP, and SRF in WI-38 cells. Interestingly, in C2C12 myoblasts, there were no effects on RhoA and MRTF-B from the CCG drugs treatment, while both CCG molecules downregulated MRTF-A. Both SRF and cofactor p49/STRAP were also downregulated (Appendix A). These results suggest that both CCG molecules act through the disruption of Rho/MRTF/SRF transcriptional signaling, and their mechanism of action is specific to cell types. 

### 3.3. CCG-203971 and CCG-232601 Induce Histone H4 Hyperacetylation 

Histone H4 acetylation is a key epigenetic marker involved in gene regulation, DNA repair, and chromatin remodeling. It also regulates the SRF association with CArG box DNA [33]. Therefore, we explored the effect of CCG molecules on histone H4 acetylation. WI-38 cells were treated with CCG molecules at 20 µM for 24 h. We performed immunoprecipitation to isolate histone H4. No change for histones H4K5ac and H4K8ac were found in treated cells versus control (Figure 3A). Interestingly, we found CCG drugs hyper-acetylate histone 4 at lysine 12 and lysine 16 residues (Figure 3B). The molecular weights of hyper-acetylated H4 lysines were observed at ~50 kDa. Immunoblots demonstrating low molecular weight of acetylated lysines at ~25 kDa are shown in Appendix A. These results suggest that hyperacetylation of histone H4K12 and H4K16 could be modulating the gene expression programs and regulating the access of the MRTFs to SRF on the treatment of CCG molecules.

### 3.4. Actin Dynamics and Mitochondrial Biogenesis Are Regulated by CCG-203971 and CCG-232601

Rho GTPases regulate SRF with their ability to induce actin polymerization and actin cytoskeleton associated with the mitochondria to regulate their dynamics and biogenesis [34,35]. Thus, we examined the influence of CCG molecules on actin assembly by staining with phalloidin, revealing alterations in the actin filaments of the treated WI-38 cells (Figure 4A–C). Similar effects were observed in C2C12 cells (Appendix A). We also examined the effect of CCG molecules on mitochondrial membrane potential (MMP) (Figure 4D–G) and C2C12 cells (Appendix A) by MitoTracker Red CMXRos staining, which was analyzed as relative fluorescence intensity. CMXRos is a lipophilic cationic fluorescent dye that is concentrated inside mitochondria by their negative mitochondrial membrane potential. The fluorescence intensity of MitoTracker Red was decreased in CCG drugs treatment versus non-treated in both cell lines that indicate the reduction in the MMP in CCG treated cells. Further, the CCG treatments evoked the downregulation of *Fis1* and *Drp1* mitochondrial fission genes expression in WI-38 cells (Figure 5A). We also observed a significant reduction in fusion genes *Mfn1*, *Mfn2*, and *Opa1*, which control the fusion of the inner membrane (Figure 5B). Mitochondrial biogenesis is controlled by the expression of mitochondrial-encoded genes, which is orchestrated by master regulators such as *PGC-1α* and *PGC-1β*. We observed significant reduction in *PGC-1α* and *PGC-1β* gene expression, as well as downregulation in the protein expression of NT-PGC-1α, a shorter isoform of PGC-1α (Figure 5C,D).

### 3.5. CCG-203971 and CCG 232601 Induce Reduction in Mitochondrial Respiration and Switch to Non-Oxidative Cellular Bioenergetics

We examined the effect of CCG molecules on cellular bioenergetics in WI-38 fibroblast cells and C2C12 myoblasts. CCG molecules exhibited impaired mitochondrial OXPHOS and reduced mitochondrial ATP levels at different treatment dosages after 24 h. A marked reduction in basal respiration, maximal respiration, spare respiratory capacity, and ATP production was observed in WI-38 cells treated with 20 µM of CCG molecules (Figure 6A,C) as measured by OCR in the Seahorse XF Cell Mito Stress Test Kit. At a high dose of 50 µM, the mitochondrial respiration was completely shut down. This suggests that CCG-203971 and CCG-232601 regulate ATP production via OXPHOS. Supporting this, similar results were observed in C2C12 cells as measured by OCR in the Seahorse Mito Stress (Appendix A). These findings are coupled with a modest increase in glycolysis in WI-38 cells measured by ECAR. Elevated levels of the basal, compensatory glycolysis, and glyco-ATP were observed (Figure 6B,C). C2C12 cells also exhibit similar response in ECAR on the treatment (Appendix A). Data in OCR, ECAR, and ATP rate assays were normalized to the equal number of cells in different concentrations of CCG drug treatments. Increased glycolysis may be a compensatory response to the severe OXPHOS deficits, suggesting that these compounds could regulate mitochondrial function by OXPHOS inhibition. 

### 3.6. Functional Analysis of Mitochondrial Respiratory Chain 

CCG molecules have a substrate-inhibitor titration downregulation in the respiratory capacity as exhibited by the high-resolution in respiratory parameters of the electron transport system. The activity of complexes (I–IV) was reduced significantly in WI-38 cells after the treatment of 20 μM of CCG molecules as compared to controls; it was decreased by 30–40% (Figure 7A). These observations indicate that CCG molecules could inhibit the activity of mitochondrial complexes, leading to a reduced activity of oxidative phosphorylation pathway. The integrity of the mitochondrial layer was evaluated by including cytochrome c. The quantitative examinations of oxygen respiration rate in different complexes are shown in Figure 7A. This result is corroborated by the data of the OXPHOS (II–V) Western blot analysis (Figure 7B) which indicates CCG molecules inhibit the mitochondrial electron chain complex proteins specifically, thus regulating the mitochondrial function.

### 3.7. Increase in Mitochondrial ROS Level and Decrease in Mitochondrial Membrane Potential

Mitochondria produce ROS as a natural byproduct of the ETC activity. Because our data indicates that CCG molecules impair the mitochondrial function at complex levels, we measured the mitochondrial ROS production in WI-38 cells after treatment with 20 μM CCG molecules at for 24 h. We used MitoSOX Red (a fluorogenic dye) for selective detection of superoxide in the mitochondria of live cells and sorted MitoSOX positive cells using flow cytometry. Figure 8A shows significantly higher levels of mitochondrial ROS in treated cells compared to untreated. The mitochondrial membrane potential, which is an index of mitochondrial function, was also examined to evaluate the effect of CCG molecules. The detection of mitochondrial membrane potential in CCG treated/untreated WI-38 cells was conducted using JC−1 dye. JC−1 aggregates in mitochondria due to electronegative environment and fluoresces a red color, but at depolarization of the mitochondrial membrane, JC − I remains in monomeric form and fluoresces a green color. FACS analysis showed that after 24 h of treatment with 20 μM CCG molecules, there is significant reduction in ATP because the percentage of JC−1 monomers were increased significantly (Figure 8B). Our data indicate that the treatment of CCG molecules lead to elevated intracellular ROS and total reduction in ATP that can induce oxidative stress and apoptosis. 

## 4. Discussion

The pleotropic roles of the RhoA signaling pathway through direct interaction with downstream effectors like MRTFs, from signal transduction and to transcriptional regulation, have been extensively investigated. The breakthrough findings of the importance of RhoA/MRTFs/SRF transcription in cardiovascular, neurodegenerative, and intestinal diseases and fibrosis to cancer tumorigenesis and metastasis have been much appreciated [36,37,38,39]. Research in cancer therapeutics has highlighted the importance of molecularly targeted therapies, and such inhibitors of Rho kinase are currently under clinical trials in cancer [40]. 

The discovery of a novel small-molecule inhibitor of transcriptional signaling of RhoA has led to the new class of anticancer drugs. The identification of the first-generation molecule CCG-1423 has potency and selectivity toward Rho-overexpressing and invasive cancer cell lines [41]. CCG-1423 possessed considerable potency, but also significant cytotoxicity. Therefore, a new molecule CCG-203971 was discovered. CCG-203971 has less cytotoxicity and improved potency as well as reduced solubility, lipophilicity, and molecular weight [42]. Furthermore, improvements to the series of these molecules resulted in CCG-232601, which is metabolically stable and soluble [43]. CCG molecules have been described as efficacious in various disease models. However, these compounds were originally identified in a cell-based luciferase assay that relied on MRTF/SRF-regulated gene transcription. Therefore, their precise molecular mechanism of action is not clear.

In this study, we describe the inhibition mechanism of RhoA signaling and further propose the important role of CCG molecules in inhibiting the signaling pathways that lie upstream of mitochondria, reducing the mitochondrial function and triggering cell death. We chose human lung fibroblasts, WI-38, and mouse myoblasts, C2C12 cell lines to investigate the role of CCG molecules in the normal and healthy mitochondrial function rather than utilizing cancer cells or other disease cellular models. Cancer cells have highly mutated mitochondrial genes and are characterized by high heterogeneity in metabolic activity and phenotype; some cancer cell lines have high anaerobic OXPHOS, but many lie on high aerobic OXPHOS [44,45]. Furthermore, CCG 203971 has been shown as an antifibrotic agent [46], which is why we chose WI-38 cell line to investigate in this study. Our previous findings have also emphasized the role of CCG-1423 in regulating the mitochondrial function in C2C12 cells [31]. Therefore, we utilized C2C12 in this to validate and compare the impact of CCG-203971 and CCG-232601 to CCG-1423. 

CCG-203971 and CCG-232601 are well tolerated at the relatively high doses used in our study. Both molecules cause no cytotoxicity at low doses until 20 µM when evaluated by the MTS Assay. The calculated IC_50_ data indicate both drugs are equipotent towards WI38 and C2C12 cells. Our data suggested IC_50_ of CCG-203971 (12 µM and 10.9 µM) and CCG-232601 (14.2 µM and 12.9 µM) in WI38 and C2C12 cells, respectively. The reported IC_50_ of CCG-203971 is 0.64 µM and CCG-232601 is 0.55 towards HEK293T cells evaluated in SRE Luciferase Reporter Assay by measuring cell viability using a Gly-Phe-AFC peptide [43]. The discrepancy among IC_50_ of these compounds could be due to cell specific responses. Further different cytotoxicity assays can give different IC_50_ [47]. Previous reports have also demonstrated an influence of CCG-203971 at 30 μM dosage on the proliferation of scleroderma dermal fibroblasts [48]. Our findings indicate that both CCG-203971 and CCG-232601 disrupt Rho signaling through the functional inhibition of SRF transcriptional activity. They specifically inhibit the MRTF-A and MRTF-B and p49/STRAP. Rho GTPases/MRTFs/SRF are highly conserved among different species [49,50], but we found that CCG molecules inhibit RhoA in human lung cells, not in mouse myoblasts, although it is possible that they target RhoB/RhoC in mice. In addition, both MRTFs are inhibited in WI-38 cells, but only MRTF-A is affected in C2C12 cells. Therefore, these compounds have specific targets in different cell types. 

Alterations in histone H4 acetylation have been observed in cancer. It has been shown that histone H4 present inside mitochondria binds to mitochondrial DNA and controls the mitochondrial gene expression [51,52]. We found that CCG molecules hyper-acetylate histone H4 at lysine 12 and 16, while no change was observed at acetylation of lysine 5 and 8. The expected molecular weight of acetylated and non-acetylated histone 4 lysines is ~14 kDa, but the molecular weight for hyperacetylated histone 4 lysines could be at ~35 kDa [53]. The discrepancy in the molecular sizes of acetylated H4 lysines observed in our findings could be due to their altered mobility shift on SDS page. Acetylation neutralizes the positively charged ε-amino group without significantly changing the molecular mass of the protein. This acetylation-dependent mobility shift could be explained by the increase of the net negative charge of the SDS-histone complexes [54]. Our findings show that histone H4 hyperacetylation makes cells vulnerable with the influence of the CCG molecules, thus potentially regulating the overall function of the mitochondria. Actin dynamics and SRF transcriptional activity comprise a regulatory circuit of MRTFs that control cytoskeletal and sarcomeric dynamics. We have shown the impact of the CCG molecules on actin filaments and how these extracellular signaling molecules stimulate actin alteration. Furthermore, others have shown that SRF-cofilin-actin signaling axis modulates mitochondrial function [19]. Mitochondrial bioenergetics, biosynthesis, and signaling play an important role in tumorigenesis, therefore, targeting mitochondrial metabolism has emerged as a potential target for cancer therapy. Our findings suggest that CCG molecules inhibit the mitochondrial genes involved in fusion, fission, and biogenesis. Thus, these compounds can regulate the mitochondrial dynamics, which is pivotal for the optimal function of mitochondria and to dictate the fate of cells.

Energy metabolism alterations are an emerging hallmark in cancer, and it was thought that glycolysis was enhanced and OXPHOS capacity reduced in various cancer cells (known as the Warburg effect) [55]. However, recent research has emphasized the crucial role of OXPHOS in anabolic requirements of cancer cells [56]. We have demonstrated the role of CCG-203971 and CCG-232601 in inhibiting OXPHOS and regulating the mitochondrial function. Our present findings also demonstrate that CCG-203971 and CCG-232601 are less toxic to mitochondrial function as compared to our previous findings of CCG-1423 treatment on C2C12 cells, which has a significant effect on mitochondrial function at low dose of 10 µM [31]. Our findings demonstrate that these compounds are effective in inhibiting all the complexes of mitochondrial ETC. There are not many molecules that are known to repress all the complexes of ETC. To date, metformin is a mitochondrial ETC complex I inhibitor that has been studied in multiple clinical trials as an anticancer agent [57]. Furthermore, we have demonstrated the role of CCG molecules in inducing ROS and reducing overall ATP. In general, cancer cells have higher levels of ROS because they require more energy as they grow and spread uncontrollably. Studies have implicated that the high level of ROS can lead to cellular damage and ultimately tumor suppression [58]. Our data suggest that CCG molecules could act as an inducer of ROS that could lead to apoptosis, autophagy, or necroptosis of the cancer cells. 

## 5. Conclusions

In the present study, we demonstrated the basic inhibitory mechanism of CCG-203971 and CCG-232601. We showed that these molecules specifically inhibit RhoA, MRTF-A, and MRTF-B downstream signaling that involves cofactor p49/STRAP and transcription factor SRF. In addition, we observed the histone H4 hyperacetylation at lysine 12 and 16, which can regulate gene expression. Present findings highlight the roles of CCG-203971 and CCG-232601 in targeting mitochondrial function and inducing oxidative stress in normal cell lines. Future studies are warranted for clinical development, including in vivo model systems and potential mitochondrial targeted antitumor effects in cancer cell lines and in human cancers.

## Figures and Tables

**Figure 1 cells-13-00392-f001:**
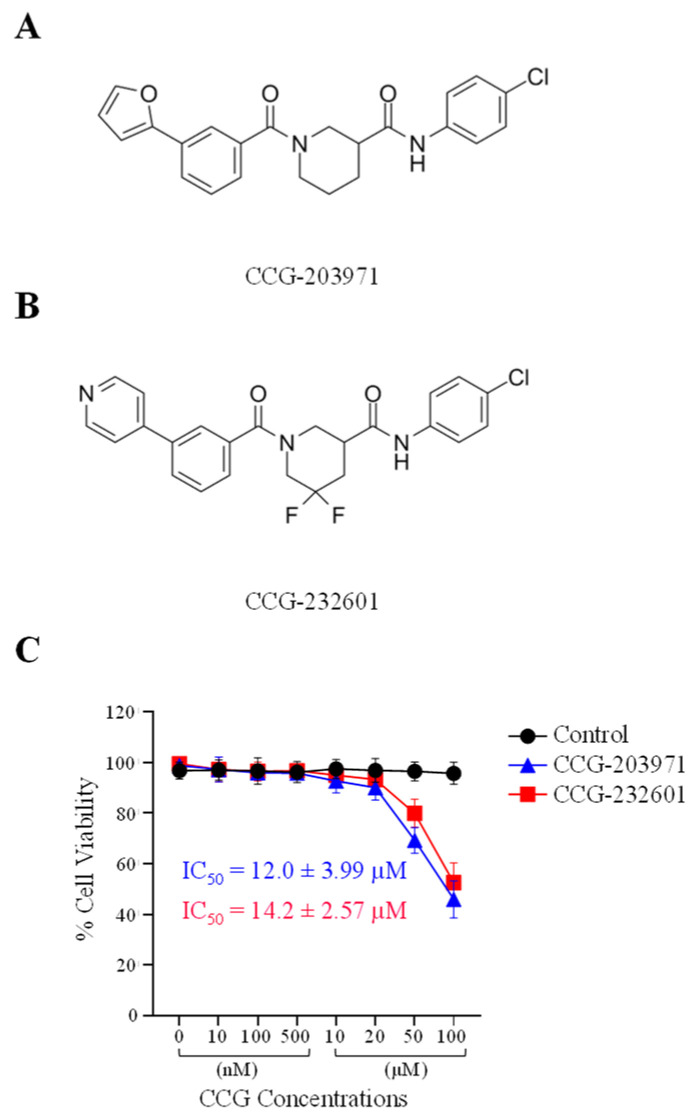
Chemical structure and cytotoxic activity of CCG molecules in WI-38 human lung fibroblasts. (**A**) CCG-203971, N-(4-Chlorophenyl)-1-[3-(2-furanyl) benzoyl]-3-piperidinecarboxamide. (**B**) CCG-232601, N-(4-chlorophenyl)-5,5-difluoro-1-[3-(4-pyridinyl) benzoyl]-3-piperidinecarboxamide. (**C**) WI-38 cells were treated with 0.5% DMSO (vehicle control, 100% viability) and increasing concentrations of CCG molecules for 24 h and subjected to MTS cell viability assay. Graphed values are shown as the mean ± SD (*n* = 4), and calculated IC_50_ values are presented as well.

**Figure 2 cells-13-00392-f002:**
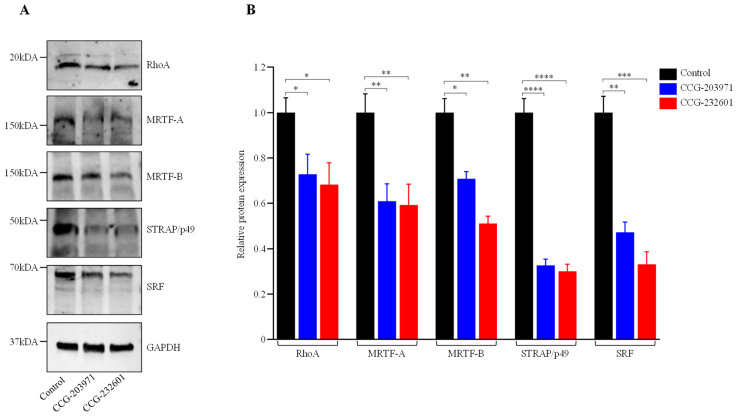
CCG-203971 and CCG-232601 inhibit Rho/MRTF/SRF signaling. WI-38 cells were treated with 20 µM of CCG molecules for 24 h. (**A**) Representative Western blots show the protein expression of RhoA, MRTF-A, MRTF-B, p49/STRAP, and SRF; GAPDH was used as a loading control. (**B**) Quantification of relative protein levels normalized against GAPDH was shown in the graphs. Data are mean ± SD of three repeats. Significant difference * *p * <  0.05; ** *p*  <  0.01; *** *p*  <  0.001; **** *p * <  0.0001 by one-way ANOVA with Tukey’s procedure.

**Figure 3 cells-13-00392-f003:**
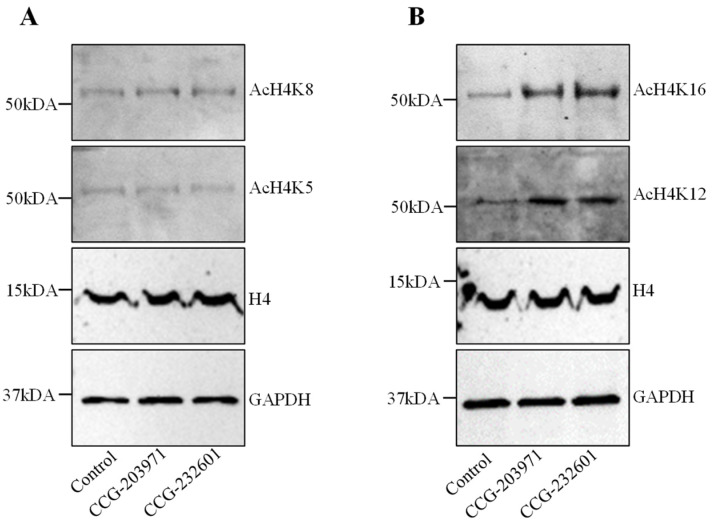
CCG-203971 and CCG-232601 regulate the histone acetylation of H4K12 and H4K16. WI-38 cells were treated with 20 µM of CCG molecules for 24 h. Western blot analysis was used to examine the expression levels of histone H4 acetylation. Total H4 and GAPDH was used as the loading control. (**A**) Representative Western blots for AcH4K5 and AcH4K8. No change was observed in the expression levels of AcH4K5 and AcH4K8 on the treatment. (**B**) Representative Western blots for AcH4K12 and AcH4K16. CCG molecules induce the hyperacetylation of H4K12 and H4K16. Blots are representative of three repeats.

**Figure 4 cells-13-00392-f004:**
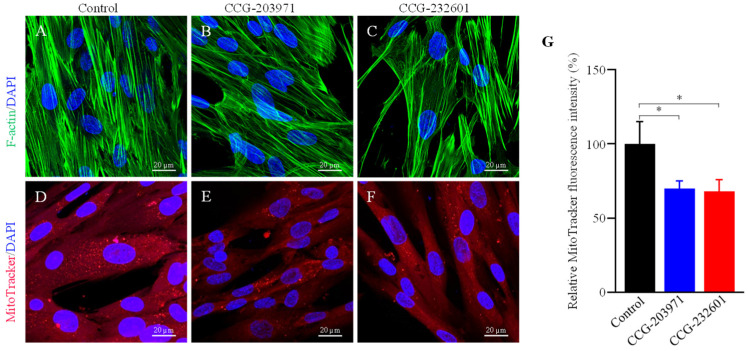
Immunofluorescence confocal microscopy images of filamentous actin. WI-38 cells were treated with 20 µM of CCG molecules for 24 h. Cells were incubated with phalloidin (green) and MitoTracker Red CMXRos conjugated dyes. DAPI (blue) was used for nuclear counterstaining. (**A**,**D**) Control (0.5% DMSO treated) (**B**,**E**) CCG-203971 treated (**C**,**F**) CCG-232601 treated. A 63× oil objective is used; scale bars indicate 20 μm. Representative images are of three independent repeats. (**G**) Bar graphs show a significant reduction in MitoTracker Red fluorescence intensity (normalized) in treated cells to control. Data plotted as mean ± SD. Data were pooled from three independent experiments (one-way ANOVA, **p* < 0.05).

**Figure 5 cells-13-00392-f005:**
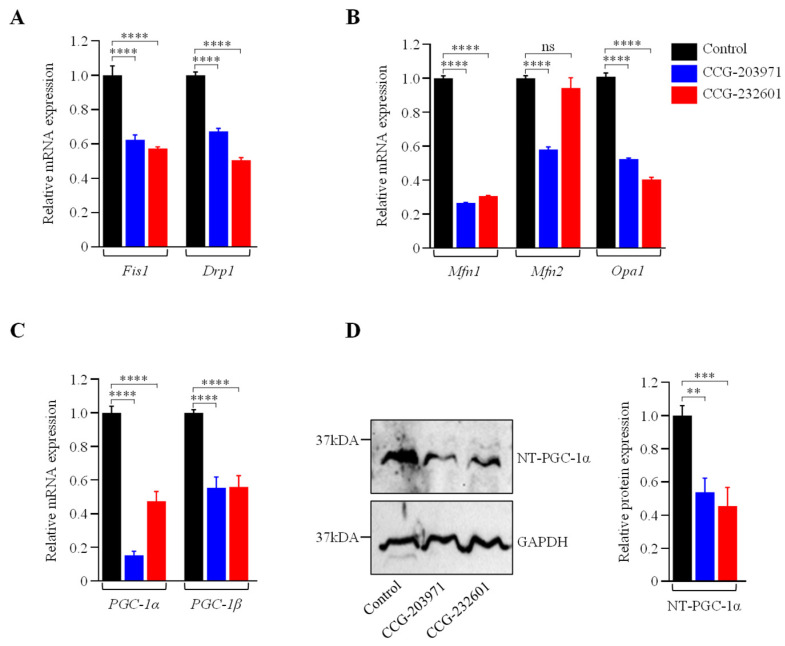
CCG-203971 and CCG-232601 repress the mitochondrial fission, fusion, and biogenesis. WI-38 cells were treated with 20 µM of CCG molecules for 24 h. (**A**) RT-qPCR analysis of mitochondrial pro-fission (*Fis1*, *Drp1*). (**B**) Pro-fusion (*Mfn1*, *Mfn2*, *Opa1*) factors show a decrease in their mRNA expression. (**C**) RT-qPCR analysis of *PGC-1α* and *PGC-1β* markers for mitochondrial biogenesis revealed downregulation of its mRNA. RT-qPCR analysis of respective genes was normalized to *5s* as a reference gene. (**D**) Representative Western blot for NT-PGC-1α and GAPDH, as a loading control. Relative quantification of protein level normalized against GAPDH is shown in the graph. Data are mean ± SD (*n* = 3). Significant difference ** *p * <  0.01; *** *p * <  0.001; **** *p * <  0.0001; ns, *p* > 0.05, by one-way ANOVA with Tukey’s procedure.

**Figure 6 cells-13-00392-f006:**
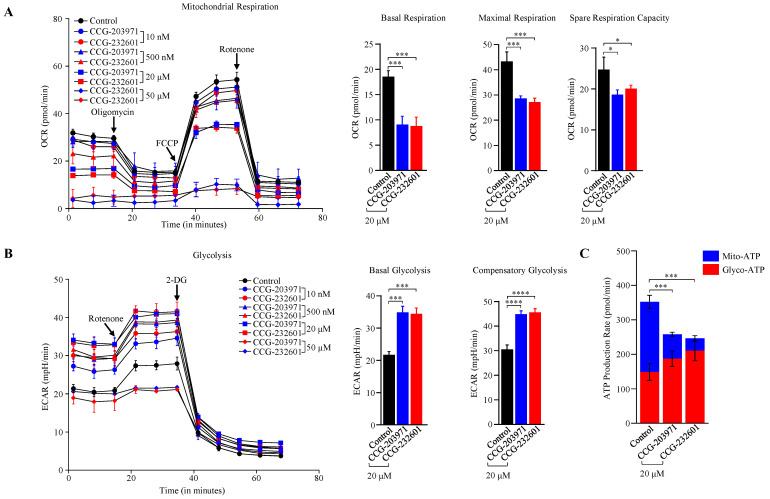
CCG-203971 and CCG-232601 regulate mitochondrial bioenergetics. WI-38 cells were treated with varying concentrations of CCG-203971 and CCG-232601 for 24 h. (**A**) Seahorse XF Cell Mito Stress Test results of control (0.5% DMSO treated) and CCG molecule-treated cells showing mean ± SD normalized to equal number of cells. Oligomycin was injected to inhibit ATP-linked mitochondrial respiration as a measure of mitochondrial ATP production. To determine maximal respiration, the mitochondrial uncoupler FCCP was injected. To determine the non-mitochondrial respiration to the OCR, rotenone was injected, inhibiting mitochondrial respiration. Quantification of basal respiration, maximal respiration, and spare respiratory capacity calculated from Seahorse XF Cell Mito Stress Test. (**B**) Glycolytic rate assay shown as mean ± SD normalized to equal number of cells. Rotenone/Antimycin A mix was injected to block mitochondrial activity and 2-DG (to inhibit glycolysis) as an internal control. Graphs show the calculated glycolytic parameters of basal and compensatory glycolysis. (**C**) Real-time ATP rate assay in WI-38 cells. On treatment with CCG molecules, there is reduction in total ATP produced from oxidative phosphorylation, while ATP generated from glycolysis was increased compared to control cells. Error bars represent mean ± SD (*n* = 3). * *p* < 0.05; *** *p* < 0.001; **** *p * <  0.0001 by one-way ANOVA with Tukey’s method.

**Figure 7 cells-13-00392-f007:**
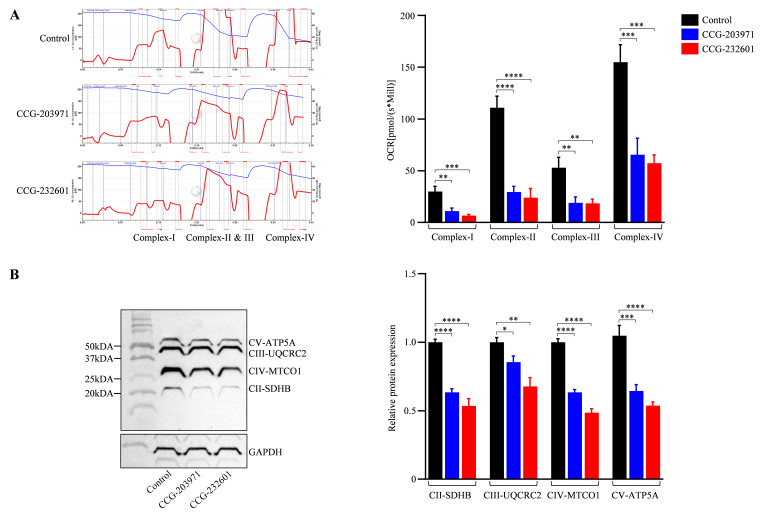
CCG-203971 and CCG-232601 inhibit mitochondrial respiratory chain complex function. WI-38 cells were treated with 20 µM of CCG molecules for 24 h. (**A**) Representative trace of high-resolution respirometry with a multiple substrate-inhibitor titration protocol. The oxygen consumption is represented as a function of time. Blue lines indicate times of titrations of substrates and inhibitors. The protocol includes the following steps: malate/glutamate (complex I-linked substrates), ADP (OXPHOS capacity), rotenone (inhibition of complex I), succinate (complex II), antimycin A (inhibition of complex III), ascorbate/TMPD (complex IV), and sodium azide (inhibition of complex IV). Quantification of complexes activity shows OCR at complex I–IV of ETC were repressed on the treatment of CCG compounds. (**B**) Representative Western blot of OXPHOS mitochondrial complexes (II-V). Antibody cocktail against complexes was used to examine the expression of mitochondrial proteins on the treatment. Relative quantification of protein levels normalized against GAPDH. Data are mean ± SD (*n* = 3). Significant difference * *p * <  0.05; ** *p * <  0.01; *** *p * <  0.001; **** *p * <  0.0001 by one-way ANOVA with Tukey’s procedure.

**Figure 8 cells-13-00392-f008:**
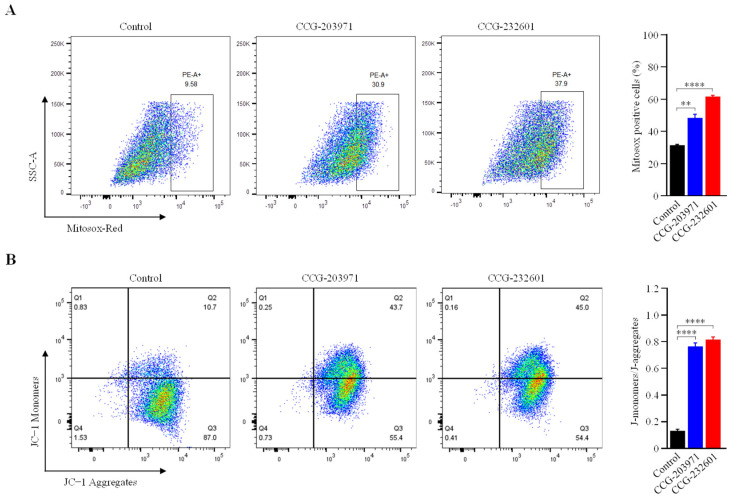
FACS analysis of JC−1 and MitoSOX red staining. WI-38 cells were treated with 20 µM of CCG molecules for 24 h. (**A**) Cells were stained with MitoSOX red, and ROS-producing cells were sorted with flow cytometry. Bar graph showing the percentage of cells positive for MitoSOX red staining. Treatment of CCG molecules induce significant ROS production. (**B**) Flow cytometry dot plot showing the gating of JC−1 (red) aggregates and JC−1 (green) monomer populations. Horizontal axis represents fluorescence intensities of JC−1 aggregates. Vertical axis represents JC−1 monomers. Increased intensity of monomers indicates decreased MMP on the CCG treatment. Graph shows ratio of JC−1 monomers and aggregates. Ratios were significantly increased in the treatment groups. Data are expressed as mean ± SD (*n* = 3). Significant difference ** *p * <  0.01; **** *p * <  0.0001 by one-way ANOVA with Tukey’s method.

## Data Availability

The data that support the findings of this study are available from the corresponding author upon reasonable request.

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
