# Peer review of "Inhibitors of Rho/MRTF/SRF Transcription Pathway Regulate Mitochondrial Function"

_cells, 2024, doi:10.3390/cells13050392_

Round 1

Reviewer 1 Report

Comments and Suggestions for Authors

The authors of this study made great advances in the mechanism of action of second generation CCG compounds and their results provide extensive knowledge in this regard. However, they mention that the compounds are selective and less cytotoxic than their parent molecule, but both compounds inhibit all mitochondrial complexes (electron transport chain) which suggests that these molecules could be disrupting the mitochondrial membrane and although  the compounds alter ATP levels do not mention anything about the phosphorylating component (complex V, adenine nucleotide translocator and the phosphate transporter). I ask the authors to please expand their explanation. 

If these molecules can alter the metabolism of normal cells (WI-38 and C2C12) how is it thought that they will not present side effects in cancer patients to be a safe treatment. please expand their explanation.

Author Response

We are appreciative of the reviewer 1 for the thoughtful comments and helpful suggestions. Please find below the responses to the reviewer’s comments.

1) The authors of this study made great advances in the mechanism of action of second-generation CCG compounds and their results provide extensive knowledge in this regard. However, they mention that the compounds are selective and less cytotoxic than their parent molecule, but both compounds inhibit all mitochondrial complexes (electron transport chain) which suggests that these molecules could be disrupting the mitochondrial membrane and although  the compounds alter ATP levels do not mention anything about the phosphorylating component (complex V, adenine nucleotide translocator and the phosphate transporter). I ask the authors to please expand their explanation. 

Response: The first-generation CCG compound, CCG-1423 and second-generation CCG compounds CCG-203971 and CCG-232601 have dose dependent effects on mitochondrial function. Our previous findings demonstrate that CCG-1423 could inhibit the mitochondrial related proteins at low dose of 10 µM (PMID: 36232837). However, second generation drugs in the current study have no effect on different mitochondrial proteins examined by western blot analysis at dosage of 10 µM. Therefore, our data suggest CCG-1423 drug is more toxic to mitochondrial function as compared to second generation compounds CCG-203971 and CCG-23260.

We did examine the inhibition of OXPHOS at protein levels. In Figure 7B, we analyzed the total protein expression by western blot analysis using OXPHOS cocktail Antibody. This antibody can detect all five complexes of ETC. In the presented figure we were able to determine expression levels of Complex V after the CCG drugs treatment. Our findings suggest that both CCG-203971 and CCG-232601 were able to repress the expression of Complex V at the protein level.  

2) If these molecules can alter the metabolism of normal cells (WI-38 and C2C12), how is it thought that they will not present side effects in cancer patients to be a safe treatment? Please expand their explanation

Response:
This is an interesting question to think about the clinical application of these drug molecules. These drugs can be utilized in Targeted Drug Therapy to treat cancers by precisely identifying and specifically targeting mitochondria of cancers cells. There have been successful targeted and precision medicine drug therapies developed to treat certain kind of cancers. However, drugs for some targets are very hard to develop due to complexity of the target’s function in the cell, further cancer cells can become resistant to targeted therapy. Therefore, more in vivo, and clinical data is required for further validation and testing of these drugs. We plan to undertake these experiments in a subsequent manuscript.

Reviewer 2 Report

Comments and Suggestions for Authors

The research conducted by Pankaj Patyal and colleagues aimed to explore the mechanism by which CCG-203971 and CCG-232601 inhibit the Rho/MRTF/SRF pathway. The introductory section is informative, scientifically sound, and comprehensible. However, the language requires substantial editing, and the results are questionable due to unscientific data interpretation. The overall research has serious flaws. Please refer to the comments below for further details.

Major comments:

1.     Several severe language issues. Such as, in lines 219-224, “decreased 70%” should be “decreased to 70%”, “CCG-223 232601 is a more potent” et al. The language must be re-edited before publishing.

2.     The data in Figure 1C and supplementary Figure 1A do not support the conclusion “CCG-232601 is more potent compared to CCG-232971”, The author should measure the IC50 and do statistics in these two cell lines instead of citing literature.

3.     By comparing the western replicates for Figure 2A, the change in the protein level of RhoA was very minor in WI-38 cells. For MRTF-A, the blot shown in the main figure is very distinguishable from repeats 2 and 3, it should be considered as an experimental error unless the author provides further verification.

4.     The molecular sizes of H4K5ac, H4K8ac, H4K12ac, and H4K16ac in Figures 3A and B are not correct. This is a failed experiment result. Please check all the data carefully before the manuscript submission.

Comments on the Quality of English Language

Needs improvement.

Author Response

We sincerely appreciate the reviewer-2 for valuable comments and suggestions, which helped us in improving the quality of the manuscript. Please find below the responses to the comments. Changes in the manuscript are highlighted in yellow.

The research conducted by Pankaj Patyal and colleagues aimed to explore the mechanism by which CCG-203971 and CCG-232601 inhibit the Rho/MRTF/SRF pathway. The introductory section is informative, scientifically sound, and comprehensible. However, the language requires substantial editing, and the results are questionable due to unscientific data interpretation. The overall research has serious flaws. Please refer to the comments below for further details.

Major comments:

1. Several severe language issues. Such as, in lines 219-224, “decreased 70%” should be “decreased to 70%”, “CCG-223 232601 is a more potent” et al. The language must be re-edited before publishing.

Response: We have reviewed and corrected the language issues throughout the manuscript.

2. The data in Figure 1C and supplementary Figure 1A do not support the conclusion “CCG-232601 is more potent compared to CCG-203971”, The author should measure the IC50 and do statistics in these two cell lines instead of citing literature.

Response: We measured the IC50 of CCG-203971 and CCG-232601 towards WI38 and C2C12 cells and found both drugs are equipotent.

We modified method section 2.3 as:

IC50 values were determined by non-linear regression analysis using GraphPad Prism 8.4.3.  

We modified both Figure 1 and Supplementary Figure 1A.

Results section 3.1 is modified as:

The data, shown as IC50 ± SD, are presented in Figure 1 and in Supplementary Figure 1A. CCG-203971 and CCG-232601 are equipotent in towards both WI-38 (IC50 = 12.0 ± 3.99 µM and 14.2 ± 2.57 µM, respectively) and C2C12 cells (IC50 = 10.9 ± 3.52 µM and 12.9 ± 2.84 µM, respectively).

We added following sentences in the Discussion part:

The calculated IC50 data indicate both drugs are equipotent towards WI38 and C2C12 cells. Our data suggested IC50 of CCG-203971 (12 µM and 10.9 µM) and CCG-232601 (14.2 µM and 12.9 µM) in WI38 and C2C12 cells respectively. The reported IC50 of CCG-203971 is 0.64 µM and CCG-232601 is 0.55 µM towards HEK293T cells evaluated in SRE Luciferase Reporter Assay by measuring cell viability using a Gly-Phe-AFC peptide [32]. The discrepancy among IC50 of these compounds could be due to cell specific response; further different cytotoxicity assays can give different IC50 [48].   

3. By comparing the western replicates for Figure 2A, the change in the protein level of RhoA was very minor in WI-38 cells. For MRTF-A, the blot shown in the main figure is very distinguishable from repeats 2 and 3; it should be considered as an experimental error unless the author provides further verification.

Response: We repeated Western blot analysis for RhoA of CCG drugs treatment (20 µM) on WI38 cells in two more replication. We quantified the relative protein expression and still find significant difference between control and treated cells. We replaced representative blot image of RhoA with the better one and added the raw files of these two replications in Western raw file document with their loading control.

We also repeated western blot analysis for MRTF-A with the treatment of CCG drugs on WI38 cells in three more replicates. We replaced better representative blot image in the main figure and raw files of three more replications are presented in the Western raw file document with their respective loading controls. We also re-quantified the relative protein expression of MRTF-A.

4. The molecular sizes of H4K5ac, H4K8ac, H4K12ac, and H4K16ac in Figures 3A and B are not correct. This is a failed experiment result. Please check all the data carefully before the manuscript submission.

Response: We fully appreciate the reviewer’s concern about molecular sizes of H4 lysine’s acetylation since histone 4 has a low molecular weight of about 11 kDa. We have carefully evaluated our findings and have addressed this issue in the following points:

1) The expected molecular weight of acetylated and non-acetylated histone 4 is around ~11-14 kDa but the molecular weight for hyperacetylated histone 4 is ~ 35 kDa. Shoulars et al., 2002 (PMID: 12207884) have shown that acetylated forms of histone H4 could be at 11- and 35-kDa.
Further, commercial antibody seller, Santa Cruz Biotech also mentioned and confirmed the molecular weight of hyper-acetylated Histone H4 at around 35 kDa. (https://www.scbt.com/p/histone-h4-antibody-f-9). In the present study, hyperacetylated H4K12, and H4K16 have molecular weights at around 35 kDa. Our previous findings with parent molecule of CCG-1423 also induced hyperacetylation of H4K16 in C2C12 cells, which has a smear band at ~35 kDa (PMID: 36232837).

2) This increased molecular weight of hyper-acetylated Histone H4 at around 35 kDa could be explained by the addition of acetyl groups to the histone protein at these lysine residues. With the treatment of these drugs, Histone H4, possibly undergoes post-translational modification with increased acetylation. This process might involve adding multiple acetyl groups to the amino acid residues of the histone 4 that would contribute to an increase in its molecular weight.

3) We used Histone H4 mouse monoclonal Ab to pull down this protein from total protein lysate using magnetic beads. The blots shown for loading control for untreated and treated cells, H4 are at expected molecular sizes of ~11-14 kDa. In addition, we have repeated these experiments in three replicates, and obtained the same results. Moreover, we also used high percentage of  SDS gels (12-15 %) to better separate these low molecular proteins. Finally, we used monoclonal antibodies against other acetyl-histone H4 lysine. Because of the several precautionary steps described above to better delineate H4 lysine acetylated bands, we believe that our data submitted for the manuscript is quite robust.

Again, thank you for all your thoughtful review, questions, and suggestions.

Reviewer 3 Report

Comments and Suggestions for Authors

The authors have presented an interesting manuscript that will interest the readers of cells. They have investigated the role of Inhibitors of Rho/Myocardin in regulating mitochondrial function and transcription. The manuscript is well-written, but some major concerns must be addressed before accepting this manuscript. These comments are meant to improve the manuscript.

Authors should explain the discrepancy in the reported literature and their data. If IC50 of CCG-232601 is 0.55 μM but authors observe almost 90% viability at 10um and the same is the case with CCG-203971.

The conclusion that there is a reduction in the mitochondrial mass is not supported by strong evidence. MitoTracker Red CMXRos is not a suitable dye for this particular experiment. This dye is a potential-based dye that tends to accumulate in active mitochondria. Perhaps using MitoTracker green or MitoTracker deep red, or immunostaining with a mitochondrial marker like Tom20 could yield better results for this experiment.

If authors want to conclude that there is a decrease in mitochondrial mass they should repeat this experiment with a different dye or a mitochondrial marker.

Mitochondrial images presented in this manuscript look hyper-fragmented in both cell lines. C2C12 is known to have elongated mitochondria.

Figure 6: This is a very nicely done figure. However, the authors should report how they normalized the data for the seahorse. Without normalization, interpreting seahorse could be very misleading.

In order to ensure the accuracy and reliability of the results, then these experiments should be repeated with normalization.

The conclusion is that these compounds specifically target OCR, not ECAR, but the evidence provided is not enough to make this conclusion. They should tone down this conclusion.

During the discussion, they claimed that the inhibitors are inhibiting all mitochondrial complexes. However, this claim could be misleading since they have not assessed the total protein levels of protein in these complexes. The observed effect could be due to transcription repression rather than direct inhibition of the complex.

Figure 7: This is a good figure

Minor remarks

F-actin phalloidin is not the correct nomenclature it should be phalloidin.

Does histone acetylation add anything to the story?

Author Response

We sincerely appreciate reviewer-3 for valuable comments and suggestions, which helped us in improving the quality of the manuscript. Please find below the responses to the comments. Changes in the manuscript are highlighted in yellow.

The authors have presented an interesting manuscript that will interest the readers of cells. They have investigated the role of Inhibitors of Rho/Myocardin in regulating mitochondrial function and transcription. The manuscript is well-written, but some major concerns must be addressed before accepting this manuscript. These comments are meant to improve the manuscript.

1) Authors should explain the discrepancy in the reported literature and their data. If IC50 of CCG-232601 is 0.55 μM but authors observe almost 90% viability at 10um and the same is the case with CCG-203971.

Response: We agree with the reviewer’s concern about the discrepancy of IC50 between previous reported and our data. We think, this variation is mostly due to differences in testing method and more importantly the type of cell lines to be used to investigate.

In the revised manuscript, we measured the IC50 of CCG-203971 and CCG-232601 towards WI38 and C2C12 cells and found both drugs are equipotent.

We modified method section 2.3 as:

IC50 values were determined by non-linear regression analysis using GraphPad Prism 8.4.3. 

We modified both Figure 1 and Supplementary Figure 1A.

Results section 3.1 is modified as:

The data, shown as IC50 ± SD, are presented in Figure 1 and in Supplementary Figure 1A. CCG-203971 and CCG-232601 are equipotent in towards both WI-38 (IC50 = 12.0 ± 3.99 µM and 14.2 ± 2.57 µM, respectively) and C2C12 cells (IC50 = 10.9 ± 3.52 µM and 12.9 ± 2.84 µM, respectively).

We added following sentences in the Discussion part:

The calculated IC50 data indicate both drugs are equipotent towards WI38 and C2C12 cells. Our data suggested IC50 of CCG-203971 (12 µM and 10.9 µM) and CCG-232601 (14.2 µM and 12.9 µM) in WI38 and C2C12 cells respectively. The reported IC50 of CCG-203971 is 0.64 µM and CCG-232601 is 0.55 µM towards HEK293T cells evaluated in SRE Luciferase Reporter Assay by measuring cell viability using a Gly-Phe-AFC peptide [32]. The discrepancy among IC50 of these compounds could be due to cell specific response; further different cytotoxicity assays can give different IC50 [48].    

2) The conclusion that there is a reduction in the mitochondrial mass is not supported by strong evidence. MitoTracker Red CMXRos is not a suitable dye for this particular experiment. This dye is a potential-based dye that tends to accumulate in active mitochondria. Perhaps using MitoTracker green or MitoTracker deep red, or immunostaining with a mitochondrial marker like Tom20 could yield better results for this experiment.
If authors want to conclude that there is a decrease in mitochondrial mass, they should repeat this experiment with a different dye or a mitochondrial marker.

Response:
We agree with the reviewer, MitoTracker Red CMXRos is not an appropriate dye to label mitochondria mass. However, we made a mistake in the writing and intended to use this dye to measure the mitochondrial membrane potential and not to conclude about mitochondrial mass. CMXRos is a lipophilic cationic fluorescent dye that is concentrated inside mitochondria by their negative mitochondrial membrane potential. We corrected this mistake and presented the relative fluorescence intensity of MitoTracker of treated versus control in both cell lines to better make the conclusion. The reduction in the fluorescence intensity of MitoTracker in CCG treated drugs demonstrate the reduction in MMP.  

We have modified results section 3.4 as:

We also examined the effect of CCG molecules on mitochondrial membrane potential (MMP) (Figures 4D, 4E, 4F, 4G) and C2C12 cells (Supplementary Figures 2D, 2E, 2F, 2G) by MitoTracker Red CMXRos staining, which was analyzed as relative fluorescence in-tensity. CMXRos is a lipophilic cationic fluorescent dye that is concentrated inside mitochondria by their negative mitochondrial membrane potential. The fluorescence in-tensity of MitoTracker Red was decreased in CCG drugs treatment versus non-treated in both cell lines that indicate the reduction in the MMP in CCG treated cells.

3) Mitochondrial images presented in this manuscript look hyper-fragmented in both cell lines. C2C12 is known to have elongated mitochondria.

Response: We have replaced the representative images of MitoTracker Staining in both cell lines.

 4) Figure 6: This is a very nicely done figure. However, the authors should report how they normalized the data for the seahorse. Without normalization, interpreting seahorse could be very misleading. In order to ensure the accuracy and reliability of the results, then these experiments should be repeated with normalization.

Response:
We presented the data of OCR, ECAR and ATP rate assay in both cell lines in Figure 6 and Supplementary Figure 3, is normalized to equal number of cells in different drug concentrations for both CCG drugs.

We had mentioned in the material and methods section 2.7 : “Measurements from OCR, ECAR, and ATP assay experiments were normalized to equal number of cells in all variables. After the experiment, seahorse plate was used to perform cell count using trypan blue exclusion method and data is normalized accordingly.”

In the figure legend we mentioned that:

“Seahorse XF Cell Mito Stress Test results of control (0.5% DMSO treated) and CCG molecule-treated cells showing mean ± SD normalized to equal number of cells”.
“Glycolytic rate assay shown as mean ± SD normalized to equal number of cells.”

5) The conclusion is that these compounds specifically target OCR, not ECAR, but the evidence provided is not enough to make this conclusion. They should tone down this conclusion.

Response:
We agree with the reviewer that we don’t have sufficient evidence to show specific inhibition of OCR and ECAR. The inhibition of OCR might have increased ECAR which is interpreted to represent compensatory flux.

We have corrected this conclusion and have rewritten it as: Increased glycolysis may be a compensatory response to the severe OXPHOS deficits suggesting these compounds could regulate mitochondrial function by OXPHOS inhibition.  

6) During the discussion, they claimed that the inhibitors are inhibiting all mitochondrial complexes. However, this claim could be misleading since they have not assessed the total protein levels of protein in these complexes. The observed effect could be due to transcription repression rather than direct inhibition of the complex.

Response: We did examine the inhibition of OXPHOS at protein levels. In Figure 7B, we analyzed the total protein expression by western blot analysis using OXPHOS cocktail Antibody. This antibody can detect all the five complexes of ETC. In the presented figure we were not able to determine the low molecular weight expression of complex I, but we were able to show  other complexes of ETC ( II, III, IV and V) are inhibited by both CCG drug molecules. Further, our Oroboros figure 7A, shows functional inhibition of Complex I to Complex IV. Oroboros protocol is limited to determine function of complex V. 

 7) Figure 7: This is a good figure.  

Minor remarks

1) F-actin phalloidin is not the correct nomenclature it should be phalloidin.

Response:
We have corrected this mistake.

2) Does histone acetylation add anything to the story?

Response:
The role of acetylation of histone H4 in epigenetic regulation decompact nucleosome and regulate the gene expression programs. Histone H4 acetylation regulates the SRF association with CArG box DNA. With the treatment of CCG drugs, histone acetylation at H4K16 and H4K12 possibly regulating the Rho/MRTF/SRF transcription, which in turn can regulate other genes possibly mitochondrial genes involve in biogenesis and function.

Round 2

Reviewer 2 Report

Comments and Suggestions for Authors

Regarding the H4 acetylation, to be open-minded and precise, I still have some concerns:

1.     It’s nothing new that histones appear at high molecular weight (~22kDa, 35kDa) due to the adding of huge modification groups such as ubiquitination (~8.5kDa). Usually, these high molecular weight histones are very rare compared to regular histones. If the western blot signals of H4 acetylations are real in this paper, I would expect to see a huge H4 acetylation signal at ~13kDa, but a weak signal at higher molecular weight. However, in this research, no H4 acetylation signal was detected at ~13kDa.

2.     The acetyl group is very small ~0.042kDa, even with multiple acetylations, it won’t affect the molecular weight that much.

3.     H4 antibody can detect the total H4 protein. Why hyperacetylated H4 bands (25kDa, 37kDa) didn’t show up in the H4 blot?

4.     Why did those H4K5ac, H4K8ac, H4K12ac, and H4K16ac antibodies used in this research only detect hyperacetylated H4? Why these antibodies didn’t detect any regular acetylated H4 at ~13kDa? It’s impossible that the cell does not have any regular acetylated H4.

5.     Looking at the H4 blot, 99% of H4 was at ~13kDa, but why none of the H4 at ~13kDa was acetylated?

6.     Lines 294-295, the researcher mentions they performed “immunoprecipitation” to isolate histone H4, and then performed western blot to analyze H4 acetylations. However, the methods section “2.6. Western blot analysis and immunoprecipitation” did not contain any detail about immunoprecipitation. The cited paper [29] also didn’t have immunoprecipitation methods. Please provide details about how the immunoprecipitation assay was performed. Such as, which antibody and how much was used for immunoprecipitation? For western blot, what was used for the denaturing of samples?

7.     Regarding whether the bands ~25kDa ~35kDa are hyperacetylated histone H4, If the researchers use these H4 acetylation antibodies to immunoprecipitate proteins, use Coomassie or silver staining show the proteins are in the ~25kDa to ~35kDa range, and subsequently analyze the ~35kDa band using mass spectrometry, confirming it as histone H4, then I would be convinced and appreciative of these findings.

Author Response

Regarding the H4 acetylation, to be open-minded and precise, I still have some concerns:

Reviewer has asked valid questions, and we are hoping to convince the reviewer about our findings of Figure 2. Please see below responses to the raised concerns:

1. It’s nothing new that histones appear at high molecular weight (~22kDa, 35kDa) due to the adding of huge modification groups such as ubiquitination (~8.5kDa). Usually, these high molecular weight histones are very rare compared to regular histones. If the western blot signals of H4 acetylations are real in this paper, I would expect to see a huge H4 acetylation signal at ~13kDa, but a weak signal at higher molecular weight. However, in this research, no H4 acetylation signal was detected at ~13kDa.

Response: We agree that ubiquitination could result in high histone 4 molecular weight. However, Histone 4 acetylation can also lead to a high molecular weight of histone 4 that is close to 35kDa.

Yes, these events of high molecular weight histone acetylation are rare and there are few publications that use the term hyper-acetylation of histone 4 but sadly molecular weights of those published papers are not shown so it is difficult for us to interpret their findings and to cite them.

But please check Figure 3 of PMID: 12207884. This paper shows histone 4 acetylated at lysine-5, -8, -12 and -16 can have a single band of 35 kDa or it can have both bands at 35 kDa and 11 kDa. They confirmed these bands of H4 lysine acetylation by amino acid sequencing. Therefore, it is not necessary to have a big band at 11-13 kDa for histone 4 lysine hyper-acetylation. If it does, then one can conclude, there is increase in acetylation but since we observed increase in band size in treated cells at ~37 KDa, therefore we believe it is hyper-acetylated.

Now, in our raw blots of all H4K5ac, H4K8ac, H4K12ac, H4K16ac, if we see carefully there are two sets of bands one at ~37KDa and one at ~ 20KDa. We think, these are the two sizes of acetylated histone 4, that are actual bands, and these bands should be at ~35 KDa and ~11 KDa. But they might have altered mobility on SDS page. This altered migration of histone 4 acetylated proteins is also documented (PMID: 10222016). Acetylation neutralizes the positively charged ε-amino group without significantly changing the molecular mass of the protein the acetylation-dependent mobility shift could be explained by the increase of the net negative charge of the SDS-histone complexes.

Furthermore, in our H4K12ac and H4K16ac raw blots treated cells have increased histone 4 acetylation both at upper band of ~37KDa and also at lower band at ~ 20KDa.

Please note that difference in migration of bands could also be cell-type specific, and as long as the results are consistent for a specific cell-type, they need to be reported. Please also note that there is actually H4 acetylation band, which supposed to be ~11-14 kDa but due to altered migration shift, it is at ~ 20KDa and this finding is consistent in our replications. 

2. The acetyl group is very small ~0.042kDa, even with multiple acetylations, it won’t affect the molecular weight that much.

Response:
Thank you for pointing this out. Yes, theoretically acetyl group is very small but practically researchers have proved histone 4 acetylation at lysine can have high molecular weight at 35 KDa. Again, please check this paper (PMID: 12207884) from the lab of Dr. Barry M Markaverich from Baylor, TX.

3. H4 antibody can detect the total H4 protein. Why hyperacetylated H4 bands (25kDa, 37kDa) didn’t show up in the H4 blot?

Response:
The Antibody that we used is monoclonal anti-H4 antibody not anti-acetyl-H4. Hence, it would not detect any acetylated bands.

We used H4 (non-acetylated, monoclonal antibody) blots to show that our total protein lysate samples (control versus treated) have equal amount of H4 and we have loaded equal amount of protein for each sample and these H4 blots served as loading controls.

4. Why did those H4K5ac, H4K8ac, H4K12ac, and H4K16ac antibodies used in this research only detect hyperacetylated H4? Why these antibodies didn’t detect any regular acetylated H4 at ~13kDa? It’s impossible that the cell does not have any regular acetylated H4.

Response: We explained this issue in comment #1, as altered mobility shift. We do have regular acetylated H4 at ~13kDa, but due to altered mobility shift of acetylated histone 4 lysine we observed them at ~20 kDa. Apart from altered mobility shift, this migration to 20kDa might be cell-type specific. Our results are consistent.

5. Looking at the H4 blot, 99% of H4 was at ~13kDa, but why none of the H4 at ~13kDa was acetylated?

Response:
Because antibody that we used is monoclonal anti-H4 not anti-acetyl H4. Therefore, we do not expect any acetylated band in H4 blot.

5. Lines 294-295, the researcher mentions they performed “immunoprecipitation” to isolate histone H4, and then performed western blot to analyze H4 acetylations. However, the methods section “2.6. Western blot analysis and immunoprecipitation” did not contain any detail about immunoprecipitation. The cited paper [29] also didn’t have immunoprecipitation methods. Please provide details about how the immunoprecipitation assay was performed. Such as, which antibody and how much was used for immunoprecipitation? For western blot, what was used for the denaturing of samples?

Response:
We have modified the materials and methods section 2.6 and discussed the protocol in detail.

The following information has been added in section 2.6 and highlighted in green color in the manuscript.

“For immunoprecipitation (IP), dynabeads were conjugated with Histone H4 antibody (1 µg/sample) for 1 hour followed by washing of beads 3 times with RIPA lysis buffer. Lysates protein (500 µg/sample) from treated or control cells were added into Histone H4 conjugated beads and incubated using a shaker at 4°C for 1 hour. Beads were then washed 3 times with same lysis buffer and resuspended in 50 μL of 4× sample buffer, followed by boiling for 10 min and separation by SDS-PAGE (15 %). IP blots were developed against anti-acetyl-H4K5, anti-acetyl-H4K8, anti-acetyl-H4K12, and anti-acetyl-H4K16. For loading control H4, 20 µg protein was denatured and loaded on SDS-PAGE from total lysate.”

5. Regarding whether the bands ~25kDa ~35kDa are hyperacetylated histone H4, If the researchers use these H4 acetylation antibodies to immunoprecipitate proteins, use Coomassie or silver staining show the proteins are in the ~25kDa to ~35kDa range, and subsequently analyze the ~35kDa band using mass spectrometry, confirming it as histone H4, then I would be convinced and appreciative of these findings.

Response:
We agree with the reviewer that we can confirm these bands by utilizing mass spectrometry, but since this was not the main focus of our experiments and it only adds information that these CCG inhibitors could change epigenetic modification of Histone 4 at K12 and K16, we will consider this approach in a subsequent study.

We very much appreciate the reviewer’s insights and depth of discussion.

On other note, if reviewer still not convinced, we could potentially remove Figure 2 because the conclusions of our manuscript wouldn’t be affected without Figure 2. Please advise.

Reviewer 3 Report

Comments and Suggestions for Authors

The authors have addressed most of the concerns I had with the manuscript, and this can be accepted in the current form.

Author Response

We are thankful to reviewer-3 for endorsing our manuscript for the acceptance.